# Fusion of the SLAM with Wi-Fi-Based Positioning Methods for Mobile Robot-Based Learning Data Collection, Localization, and Tracking in Indoor Spaces

**DOI:** 10.3390/s20185182

**Published:** 2020-09-11

**Authors:** Gunwoo Lee, Byeong-Cheol Moon, Sangjae Lee, Dongsoo Han

**Affiliations:** Department of Computer Science, Korea Advanced Institute of Science and Technology (KAIST), Daejeon 34141, Korea; gwleee@kaist.ac.kr (G.L.); chul7672@kaist.ac.kr (B.-C.M.); summit@kaist.ac.kr (S.L.)

**Keywords:** mobile robot, indoor localization, learning data collection

## Abstract

The ability to estimate the current locations of mobile robots that move in a limited workspace and perform tasks is fundamental in robotic services. However, even if the robot is given a map of the workspace, it is not easy to quickly and accurately determine its own location by relying only on dead reckoning. In this paper, a new signal fluctuation matrix and a tracking algorithm that combines the extended Viterbi algorithm and odometer information are proposed to improve the accuracy of robot location tracking. In addition, to collect high-quality learning data, we introduce a fusion method called simultaneous localization and mapping and Wi-Fi fingerprinting techniques. The results of the experiments conducted in an office environment confirm that the proposed methods provide accurate and efficient tracking results. We hope that the proposed methods will also be applied to different fields, such as the Internet of Things, to support real-life activities.

## 1. Introduction

With the recent rapid developments in the field of artificial intelligence, research on robots has become more active; such robots are being used in various service fields, such as robot vacuum cleaners. Robot localization is an essential technology used in robots as a service (RaaS). The global positioning system (GPS) can be used to estimate the robot’s location outdoors. However, GPS signals become unavailable due to signal blocking in most indoor environments [1], making it difficult to accurately locate mobile robots. To overcome these limitations, researchers have developed simultaneous localization and mapping (SLAM) technology to allow the robot to create a map and use it to determine its location and focused on further improving this technology [2,3,4]. However, in general, the robot SLAM technology has uncertainty in the location recognition and control of self-propelled robots due to physical influences, such as the precision of the odometer sensor of the robot and external impact. In addition, if a new hardware is installed, then the scalability problem may rise because of the high cost. The errors caused by sensors used in SLAM are also an issue to be resolved in robot positioning.

As an alternative for solving the uncertainty of SLAM technology and for a more precise robot location recognition and control, Wi-Fi-based positioning, which is widely used in pedestrian localization [5,6,7], has attracted attention. Unlike wireless technologies, such as Bluetooth, which require additional infrastructure construction, Wi-Fi is already installed in many residential or commercial spaces, so no additional installation is required. In Wi-Fi-based positioning, Wi-Fi fingerprinting technology collects and uses the strength of Wi-Fi signals at a specific location [8,9]. Wi-Fi fingerprinting technology utilizes the received signal strength (RSS) of Wi-Fi signals collected in an indoor space. To do this task, a Wi-Fi radio map (WRM), a set of RSS measurements that knows the location, must be built in advance, and the location is estimated by comparing the WRM with the collected user RSS values [10].

In the Wi-Fi-based localization technique, even if the same algorithm is used in the same place, the positioning accuracy greatly varies depending on the signal environment and the precision of the built WRM. Since fingerprints are unique characteristics of signals for each space, the more they are densely collected, the more accurate the position evaluation [11]. In addition, as more data are collected in an environment in which a plurality of access points (APs) exist, a high-accuracy positioning result can be derived by accurately modeling the signal intensity distribution at a corresponding location. However, a dedicated user must be manually involved in collecting high-density, large-capacity data, and considerable time and effort are required in the data collection. Various studies have been conducted to lower the data collection cost [1,12,13], but more efforts are required to increase the reliability of the data quality.

SLAM-based radio map construction is a post-processing of the collected data while moving in an arbitrary path, unlike humans directly making a WRM construction plan. Therefore, there may be cases in which data collected in a specific place are biased or not collected. In this case, a deviation of a localization accuracy may occur between an area in which a large number of training data is collected and an area in which the plurality of the training data is not collected. In addition, a phenomenon in which the positioning results are biased to the area where several data are collected may occur. Therefore, to build a high-quality WRM that enables accurate positioning services, it is necessary to check whether or not the appropriate level of learning data is collected through the analysis of the learning data.

To solve these problems, we introduce a SLAM-based data collection and analysis (SDCA) technique that analyzes the reliability of WRM and learning data collected by robot SLAM and plans/executes the additional collection of learning data. In addition, we propose the extended Viterbi algorithm and signal fluctuation matrix fusion tracking method (EVSFM), which is a technique for recognizing the robot location through the interworking of Wi-Fi-based positioning and SLAM. SDCA determines the validity of the Wi-Fi signal data obtained from a robot equipped with a SLAM module and processes and trains it to construct and manage a WRM. In EVSFM, using the WRM built with SDCA, the extended Viterbi algorithm, signal similarity matrix, and odometer information of the robot are fused to perform localization and tracking. An experiment was conducted on an actual building to verify the SDCA and EVSFM. The proposed SDCA method achieved 12.9% improvement of the WRM construction, and the EVSFM achieved 40% improvement compared to the deterministic method, such as the k-nearest neighbors (kNN) algorithm.

The remainder of this paper is organized as follows: Section 2 introduces related research on robot localization methods. Section 3 presents the learning data collection method using SDCA. Section 4 describes the WRM construction of the proposed SDCA and WRM optimization. Section 5 introduces the proposed method EVSFM. Section 6 presents and analyzes the evaluation results obtained for the proposed system. Finally, Section 7 concludes this paper. 

## 2. Related Works

Research using robot SLAM and Wi-Fi-based localization has been conducted in various fields, particularly in robot positioning and tracking. Although fewer studies were simultaneously conducted using robot SLAM and Wi-Fi-based localization, the results of the research using each method show the lack of applications in the field of indoor robot localization. Li et al. [14] proposed a hybrid intelligent system of a human and robot for exploring an unknown environment. The proposed system uses object detection-based deep learning and improved SLAM. They selected feature points extracted from images and then reduced localization errors caused by the moving objects. However, the deep-learning object recognition-based feature point selection requires retraining, which may limit its general applicability.

Chen et al. [15] made a strategy that tightly couples a stereo camera and inertial measurement unit (IMU) for a better estimation of the poses of a mobile robot. When the robot is working, the forward–backward optical flow is used to track image features. However, this strategy may not work when the robot is operated in dark and narrow environments. Guan et al. [16] designed a visible light communication localization and navigation package based on the distributed framework of the robot operating system (ROS), which contains the basic operation control of the robots, LED identification detection, and dynamic cm-level visible light positioning algorithms for robot indoor localization. Lv et al. [17] presented a novel indoor localization method for a skid-steering mobile robot by fusing readings from an encoder, gyroscope, and magnetometer, which can be read as an enhanced dead-reckoning localization method. The proposed strategy mainly consists of an orientation algorithm and a localization algorithm. However, the bias, fluctuation, and accumulated errors of each sensor were not considered in depth.

Fu et al. [18] proposed a strategy mainly consisting of an orientation algorithm and a localization algorithm. They used a SLAM system by combining the points and line information and introduced a novel pose solver to handle point and line correspondences. In addition, a represented unified optimization model concurrently minimizes the point and line reprojection errors. Nevertheless, an efficient representation of the point and the advantage of a line feature could be exploited. Faigl et al. [19] proposed a unifying approach that plans data collection missions using a multigoal path planning framework. The particular class of data collection tasks consisted of problems where it is necessary to consider penalties on unvisited sensors and the sensors are surrounded by neighborhoods. However, if appropriate penalties according to the robot travel cost are not set, then the system performance may degrade.

Ren et al. [20] adopted a matrix completion technology to recover missing data samples with partial data while guaranteeing the quality of service of the task. By focusing on the correlation of the data, the proposed method proceeds data selection in terms of the cooperation effect of the reporters rather than a single data sample. However, it is difficult to expect a good performance of the system if sufficient data are not collected in the data selection. Moreover, there is a study on robot data collection using the characteristics of a wireless sensor network (WSN). Chen et al. [21] used the threshold of the sensor node that results in the partitioned/islanded WSN. By shuttling the mobile robot to the desired locations, the lifetime of the sensor nodes could be prolonged. To minimize the total cost of the system, Yan et al. [22] considered a strategy that combines clustering and mobile robot collection.

Several researchers have also conducted indoor localization using robot SLAM and other techniques. Matellan et al. [23] presented a probabilistic localization algorithm that uses a Wi-Fi network and odometrical encoder data. In this study, the authors showed that the power signal received from the AP can be exploited without any a priori energy map. By using a breakpoint model of indoor radio propagation, they tried to reduce the use of additional information apart from the locations of the AP. They also tested the feasibility of a sampling method to reduce the computational cost of the localization algorithm. Liu et al. [24] fused the pedestrian dead reckoning (PDR) and RSS measurements from a surrounding AP. By using the graph SLAM technique and collaborative fashion, they estimated the trajectory of multiple users in an unknown environment. However, if the problem of accumulated errors of sensors used in PDR and SLAM is not clearly resolved, then it is difficult to obtain good tracking results.

## 3. Methods 

The technology and driving mechanism that constitute the robot-based positioning system consist of data collection, WRM construction, and localization steps, including robot tracking, and the detailed technological development steps are as follows:
Data collection: The Wi-Fi fingerprints are collected from the coordinates generated by the GMapping SLAM of the robot, and its purpose is to acquire high-quality learning data. The details of GMapping SLAM are beyond the scope of this study.WRM construction: WRM is created by training the acquired data when the data collection is complete.Localization: The Wi-Fi fingerprints measured by the robot are compared to the trained WRM to perform positioning.

If sufficient data is not collected in the data collection step, it becomes difficult for the system to achieve high accuracy. Collecting sufficient data is not only applicable to robot-based positioning but to all fields using learning data. The manual collection of learning data takes a lot of time and effort; therefore, walking surveys or crowdsourcing-based data collection techniques are used. However, this study deals with robot-based learning data collection, which needs uniform data without any errors.

In the WRM construction step, a database is built by mapping the collected data and location coordinates. As the performance of the positioning system depends on the quality of the WRM, WRM construction plays a very important part in the overall process. The map generated by the robot during data collection may differ from the actual map due to errors such as sensor bias or fluctuation. In addition, even if a large amount of data is collected according correctly, data shortage may occur depending on the region. To solve this problem, we use indoor maps and interpolation techniques in WRM construction for improving the positioning accuracy.

In the localization step, the location of the object is estimated using a positioning algorithm, wherein the positioning accuracy varies depending on the algorithm used. We applied a probability-based tracking method to improve the positioning accuracy.

When the basic technology in all steps is completed, the basic conditions for robot positioning are completed. The three steps, i.e., data collection, WRM construction, and localization, are independent of one another, and the technologies used in each step can also be independently advanced. In this study, advancement was performed in each step to achieve a high positioning accuracy.

Figure 1 shows the overall structure of a robot positioning system by integrating each step-by-step technology. The positioning application operates semi-independently from the robot’s SLAM node and is driven by subscribing to the topic of the SLAM node through the ROS. Each step developed as a separate module was sequentially executed, and these modules were integrated into a simple positioning/visualization application. In Section 3, the data collection step is described, and in Section 4 and Section 5, the WRM construction and localization steps are described, respectively.

### Data Colleciton of the SDCA

The learning data required for WRM construction are the Wi-Fi signals and location coordinates where the data were collected. The location of the Wi-Fi signal required for the WRM construction is the estimated value generated by the robot’s SLAM. The Wi-Fi signal indicates the Wi-Fi data that are continuously measured through the wireless local area network (WLAN) card of the laptop operating the mobile robot during SLAM. The measurement period of the laptop’s Wi-Fi signal is about 3 to 4 s, which is relatively slower than the period of updating the estimated position in the robot. Therefore, the WRM construction maps the estimated location of the robot according to the time when the Wi-Fi signal data are updated.

The analysis of the uniformity of a WRM is possible through the distribution of coordinates on the metric map of the collected training data. However, considering that the signal environment for each point is different, the analysis of the distribution of the signal data collected at each point, along with the coordinates, is more effective in improving the reliability of a WRM and determining the validity of the data. This process can be performed through methods such as a similarity analysis between adjacent signal data and cross-validation using a plurality of WRMs constructed in the same space. Furthermore, the additionally collected training data can be used to replace the training data of the existing WRM or be added to the existing data to improve the reliability of the WRM.

Figure 2 shows the results of the WRM construction as the robot builds a map on the test bed as a heat map. The darker the shade of blue, the lower the density of the data collection results, whereas the brighter the shade of yellow, the denser the data collected. As shown in the figure, the density of the collected data in the yellow-boxed area is higher than that of the red-boxed area. Since the trace of the robot is set for SLAM and not for WRM construction, the density of the WLAN fingerprints may not be uniform. However, the data were not collected at a uniform rate, which may have caused a decrease in the positioning accuracy and decreased the reliability of the training data. Therefore, we executed a Wi-Fi fingerprint recollection at certain locations where enough data were not collected.

In the recollection of Wi-Fi fingerprints, first, the map file and WRM constructed by the robot were analyzed to confirm the recollection region of the Wi-Fi fingerprints. The region where the Wi-Fi fingerprints were collected was divided into a 50-cm × 50-cm-square area limited to the area in which the robot can move. In addition, the number of Wi-Fi fingerprint collections for each rectangular area is displayed. The Wi-Fi fingerprint recollection points were selected only in the lower 30% of the total number of Wi-Fi fingerprint collections. Second, an optimal path was designed to visit all the given recollection points. The algorithm used to find the optimal path is the 2-Nearest Neighbor (2-NN) algorithm used in the traveling salesman problem. In this case, the moving distance between each Wi-Fi fingerprint collection point is not a simple linear distance. In other words, the A* algorithm was used to reflect all wall information between the collection points and used as a moving distance value similar to the actual one. 

Finally, the robot collected Wi-Fi data along the planned recollection path. The robot received the Wi-Fi fingerprint recollection information and optimal path information obtained from the previous steps and saved it as a file. This file was read from the ROS package, and the Wi-Fi fingerprints were recollected by sequentially visiting the waypoints. Figure 3 shows a heat map of the WRM after the recollection of the Wi-Fi fingerprints. As shown in the figure, most of the Wi-Fi fingerprints that were not collected in the initial WRM were supplemented from the recollection of the Wi-Fi fingerprints. By configuring the WRM that guarantees the number of Wi-Fi collections above a certain level, the overall accuracy of the Wi-Fi positioning can be improved.

## 4. WRM Construction Using SDCA

### 4.1. Indoor Space Information-Based WRM Construction

The occupancy grid map (OGM) generated from the GMapping SLAM, as presented in Section 3, contained errors owing to the sensor performance and type of the robot. Such errors can distort the ratio, distance, or angle of the OGM differently from the actual one. Figure 4a shows the results of the OGM generation on the test bed, and Figure 4b shows the corresponding area on the indoor drawing. In the part marked with a red box in Figure 4a, the angle is bent differently from the actual due to the error caused by the odometer. In the blue box, the area outside the building was incorrectly recognized as a movable structure due to an error in the laser sensor.

As a result, a significant distortion occurred when the OGM was generated in a specific structure, including a glass wall, and the generated WRM also contained the same coordinate error. Therefore, in this study, the WRM distortion was corrected, and the OGM was mapped to other indoor maps to complete an automatic coordinate conversion module for different maps (coordinate systems). When the coordinate conversion system was completed, the WRM data generated by the robot were converted into a premanufactured map coordinate system, enabling positioning in various Internet of Things (IoT) devices (e.g., smartphones).

In the first step of the map correction, the OGM and interior drawing were arranged to overlap. Figure 5a shows the indoor space of an apartment and the appearances when the collected OGM and WRM were stacked using the correction tool. The WRM correction tool basically adjusts the ratio of each image and the rotation angle of the location so that the coordinates can be unified. In other words, it is converted into WRM coordinates so that the indoor map, not the OGM, can be used. For this purpose, each transformation of the WRM coordinates according to the movement, rotation, and size adjustment is applied. This process is possible by simply multiplying each matrix as shown in Table 1.

However, as shown in Figure 5b, correction for the partial region distortion is required. The basic principle for this is the same as that in the whole conversion formula, except that it applies only to parts. Following this basic principle, in this study, the user can freely transform the distorted part by selecting it. Figure 6 shows the distorted region corrected using the transformation formula.

### 4.2. WRM Optimization

#### 4.2.1. Interpolation-Based WRM Optimization

The WRM must contain the Wi-Fi fingerprints for all locations in the target space. However, it is difficult to obtain the data of all locations when collecting training data based on SLAM. Therefore, to construct a WRM capable of producing a more accurate positioning from the collected training data, data interpolation for uncollected points is essential.

When the general interpolation method is applied to the WRM construction, many factors can cause the performance degradation of the WRM, but we focused on two problems, i.e., the signal discontinuity problem and head-cut problem. The signal discontinuity problem arises when the signal rapidly changes as it passes through a wall. Existing interpolation methods do not reflect this phenomenon and generate inaccurate WRM. Inverse distance weighting (IDW) interpolation [25] is suitable for WRM construction due to the characteristic of referencing measured values at similar points in the environment. However, even if it is an adjacent point, it is difficult to say that it is a similar environment if the adjacent points are blocked by a wall. We modified and used IDW to take advantage of being able to refer to various sensing information in robot positioning. The modified IDW does not refer to the actual measurement point blocked by the wall for interpolation. Compared to the existing IDW, it has the advantage of better reflecting the characteristics of the RSS value change due to the environment. The newly developed interpolation method uses the wall information shown in the OGM based on the existing IDW interpolation method.

Meanwhile, the head-cut problem is a problem that occurs when learning data cannot be collected at a location close to the AP and is a problem of interpolating the location RSS near the AP to a lower RSS than the actual one. In fact, the closer the location to the AP, the stronger the RSS is exponentially, and when the WRM contains these characteristics well, a high accuracy can be secured at the locations around the AP. To solve the head-cut problem, we have developed a Voronoi tessellation-based interpolation technique and applied it to the mobile environment [26]. This technique estimates the location of the AP by applying the radio signal propagation model to the training data. When interpolating the RSS of a certain point, this model infers the RSS to be interpolated from the signal propagation model that has the path-loss exponent of the nearest measured point and the estimated AP position as parameters.

#### 4.2.2. Signal Fluctuation Matrix (SFM) Construction

There are various types of ways to build an indoor space WRM. The most widely used deterministic model uses only the average value of the RSS collected at one point as the WRM, so a WRM can be constructed with little data. However, the limitation of positioning accuracy is clear. The probabilistic model also shows a higher positioning accuracy using the histogram of the actual data or the average and variance. However, relatively large amounts of data need to be collected to construct a probability model [1]. 

In addition to the existing deterministic WRM, we introduced a new WRM called SFM that records similarity scores for the fluctuations of signals, overcoming the limitations of data volume and positioning accuracy. Unlike other existing WRM models that depend on the location and APs, SFM builds a universal model for each space, so it is possible to build a reliable WRM with little data collection. In addition, using the observed value of fluctuation among the actual data, it is possible to expect high positioning accuracy by providing a fluctuation score suitable for the signal environment of the corresponding space. In an ideal environment, wireless signals follow the signal attenuation model, but fluctuations in the signal strength are frequent due to various obstacles and walls in indoor spaces, or there are cases where they are missed. Existing similarity measurement techniques simply calculate the similarity between fingerprints through numerical comparisons between numeric vectors and, thus, do not reflect the features available in fingerprint-based positioning, as described above [27]. 

The proposed SFM transforms the substitution matrix used in bioinformatics to suit the WLAN fingerprint environment. By reflecting the specificity of the radio signal through the signal variation matrix, it is possible to improve the accuracy of the similarity measurement. The substitution matrix is the most widely used technique for comparing amino acid or DNA sequences and is obtained by scoring the probability that one amino acid can be substituted for another amino acid in the evolution process. The fingerprint-based localization technique has a similarity in an environment different from the amino acid sequence, but both environments contain errors due to mutations or noises. For example, because a radio signal fingerprint collected at one place is a unique characteristic of that place, two fingerprints collected at the same place must be the same. If the signal strengths of Access Point *k* (${\mathrm{AP}}_{k}$) belonging to two fingerprints collected at the same place are different from each other, then this can be considered a fluctuation due to noise. Accordingly, a substitution matrix can be introduced for the comparison of radio signal similarity. In this case, a signal sequence may be regarded as a fingerprint, each amino acid as a signal strength, and a substitution between amino acids may be regarded as an error due to fluctuation.

In the SFM, the similarity $s_{ij}$ between different RSSs collected at the same location is expressed as follows based on the above assumption:(1)$$s_{ij}=log\left(\frac{P\left(i,\;j\right)}{P\left(i\right)P\left(j\right)}\right),\;\;i\geq j\;\left(i,\;j\in SS\right),$$
where SS is the set of receivable RSS values, and P(i, j) is the probability of fluctuations observed from the RSS pair  (i, j). P(i)P(j) is the probability of fluctuations predicted in the pair, which is the same as the probability of observing the RSS pair  (i, j) of a specific AP at one place. If the substitution between the two signal strengths occurs more frequently than expected due to signal fluctuation, then a high score is obtained. Since the signal strength follows the signal attenuation model, a higher score is obtained as the two signal strengths are similar.

In the process of generating the SFM, we confirmed that the characteristics of the signal distribution determined by the collection location of the signal and the unique characteristics of the AP are smaller than expected. In other words, if the signal strengths are similar, then they are collected at different locations, or even different APs show similar signal distributions. Focusing on this, we constructed a new type of probability model universal to each space and AP called SFM. Figure 6 shows an example of an SFM calculated through Equation (1). Each axis of the matrix represents the RSS of each fingerprint for one AP, where a blue color indicates a higher score, and a red color indicates a lower score.

In the past, positioning techniques based on probability, such as histogram and Gaussian distribution, have been studied to minimize the decrease in positioning accuracy due to the instability of Wi-Fi signals. Through a WRM construction, a large amount of training data in each point is required to build a reliable probability model. SFM builds a universal model for the AP and location, so it is possible to calculate the similarity with high accuracy while reducing the data required to build a probability model.

## 5. EVSFM

When the data collection and WRM construction are completed, the location of the robot can be estimated using a localization algorithm. This section introduces the extended Viterbi tracking algorithm, which is a modified Viterbi tracking algorithm suitable for robot tracking. In addition, the SFM-based tracking applied to the extended Viterbi algorithm and the EVSFM method using odometer information as a method to improve positioning accuracy is described.

### 5.1. Extended Viterbi Tracking

In the final step of the robot positioning, robot tracking was performed by modifying the existing Viterbi tracking algorithm. The Viterbi algorithm is a representative tracking algorithm based on the hidden Markov model (HMM). As the basic concept of HMM, which estimates the hidden state of the HMM from observation values, is similar to the indoor positioning problem, it is reasonable to use Viterbi tracking based on the HMM. However, the general Viterbi algorithm has the following difficulties.

The Viterbi algorithm needs to know the emission probability P(o|l)—that is, the probability that the fingerprint *o* will be observed in the location state *l*. To obtain this probability, a large number of training data must be collected to secure the signal distribution. That is, as the number of fingerprints collected at one location increases, the positioning result itself is accurately estimated, but the collection cost is higher in terms of time. To reduce the collection cost, the number of training data collection needs to be reduced, but the positioning accuracy decreases, so there is a trade-off relationship. Another difficulty is the calculation of the transition probability. The transition probability means the probability of moving from the current location to another location, and the transition probability is calculated using the conditional probability of the IMU sensors. Since the tracking accuracy may vary depending on the error control or probability modeling of the sensors, the calculation of the transition probability is still an issue in applying the Viterbi algorithm.

To solve these difficulties, we used a probabilistic fusion algorithm. This algorithm was created through the fusion of the algorithm obtained by developing an equation based on probability theory with the conceptual idea of the existing Viterbi algorithm. The probability used as the basis of the developed algorithm is conditional probability, which is presented as follows:(2)$$P(l_{t},l_{t-1}|o_{t},o_{t-1},d_{t,t-1},HMM)$$

As shown in Equation (2), the given model is the *HMM*, the distance traveled from t−1 to *t* is $d_{t,t-1}$, the value observed at a time t−1 is $o_{t-1}$, and the value observed at *t* is $o_{t}$. At this time, we start with finding the conditional probability that the position at t−1 is $l_{t-1}$ and the position at *t* is $l_{t}$.
(3)$$\frac{P(l_{t}|o_{t})\times P(l_{t-1}|o_{t-1})\times P(d_{t,t-1}|l_{t},l_{t-1},HMM)\times P(HMM|l_{t},l_{t-1})}{P(d_{t,t-1},HMM|o_{t},o_{t-1})}$$

By fusing the idea of the Viterbi algorithm with Equation (3), the following equation is finally derived:(4)$$V_{t}\left(o_{t},l_{t}\right)\approx P\left(l_{t}|o_{t}\right)\times \underset{l_{t-1}\in S\left(l_{t}\right)}{\max}\left[V_{t-1}\left(o_{t-1},l_{t-1}\right)\times P\left(d_{t,t-1}|l_{t},l_{t-1}\right)\right]$$

$V_{t}\left(o_{t},l_{t}\right)$ refers to the probability that the current position is $l_{t}$ when considering the observed values up to a certain time *t*, including the observed value $o_{t}$ at time *t*. The probability can be calculated using $P(l_{t}|o_{t})$, $P\left(d_{t,t-1}|l_{t},l_{t-1}\right)$, and $V_{t-1}\left(o_{t-1},l_{t-1}\right)$ already calculated at t−1. $S\left(l_{t}\right)$ means a set of all states that can be transitioned at location $l_{t}$. In the case of $P(l_{t}|o_{t})$, the probability value can be calculated using the existing similarity. The proposed algorithm solves the difficulties of the previous Viterbi algorithm as follows. First, the condition that multiple fingerprints must be collected disappears, because $P(l_{t}|o_{t})$ can be obtained based on similarity, even if only one fingerprint per location is present. In addition, the convenience and accuracy of the calculations can be ensured by explicitly calculating a probability value for a moving distance without obtaining a randomly obtained transition probability.

### 5.2. SFM Tracking

The previously constructed SFM is a model that measures the similarity between specific RSS pairs. In this study, the signal observation probability is calculated from the WRM and SFM and applied to the Viterbi-based tracking technique.

For an accurate Viterbi positioning, the score $S\left(o_{k}|l\right)$ at which the signal value $o_{k}$ of $AP_{k}$ at a given position l can be obtained using the following equation:(5)$$S\left(o_{k}|l\right)=log\left(\frac{P\left(o_{k},m_{k}\right)}{P\left(o_{k}\right)P\left(m_{k}\right)}\right)$$
where $m_{k}$ is the mean RSS of the $AP_{k}$ signals collected at location *l* recorded in the WRM. Therefore, the score S(o|l) at which a fingerprint *o* will be collected at a given position *l* can be obtained as the average of scores applied SFM for the RSS obtained from several APs as follows.
(6)$$S\left(o|l\right)=\frac{1}{n}\underset{k=1\ldots n}{\sum }S\left(o_{k}|l\right).$$

### 5.3. Using Odometer Information

In robot-based Wi-Fi localization, the moving distance and direction of the robot can be determined using odometer information. Therefore, it is possible to more accurately estimate the movement trajectory of the robot through the calculation of the transition probability using the odometer information. However, as the odometer does not have any information on the Wi-Fi positioning system, the coordinate system may have a different origin or reference axis from the positioning system. Therefore, to utilize the information of the odometer, the work of calibrating different coordinate systems has to be preceded as shown in Figure 7, where θ presents direction value, Δθ is the direction change value in the odometer, *t* presents time, *x*, *y*, Δx, and Δy denotes coordinates and its differences.

The correction operation is divided into step 1, which converts each coordinate system to a spherical coordinate system, and step 2, which calculates $\Delta \theta =\theta_{t}-\theta_{t-1}$ through the angular value $\theta_{t-1}$ at t−1.

Since the moving distance of the robot is the same in both coordinate systems, the two coordinate systems are converted into a spherical coordinate system to utilize this. This conversion is possible through the following conversion equation:(7)$$r=\sqrt{x^{2}+y^{2}},\;\theta =atan\left(\frac{y}{x}\right)$$

The converted θ value may also have a different reference axis, but as the Δθ value, which is the difference between the previous step and the current angle, is the same in both coordinate systems, it is possible to calculate a more accurate transition probability using this value. The value of θ in the Viterbi algorithm is defined as the direction between the two states. At this time, because the optimal movement trajectory at t−1 is stored for each state, the direction of the corresponding trajectory can be expressed as the direction in the t−1 step of the state. When moving from state *i* to *j*, Δθ in the Viterbi algorithm is expressed as an equation:(8)$$\Delta \theta^{t}=\theta_{j}^{t}-\theta_{i}^{t-1}=atan\left(\frac{y_{j}-y_{i}}{x_{j}-x_{i}}\right)-\theta_{i}^{t-1}$$

Assuming that the error between the above two results, *r* and Δθ, follows a Gaussian distribution, the transition probability $t_{i,j}$ from states *i* to *j* can be calculated as follows through Gaussian probability values:(9)$$t_{\mathrm{ij}}=\frac{1}{\sqrt{2\sigma_{r}^{2}\pi }}\exp\left(-\frac{{\left(r-dist\left(i,\;j\right)\right)}^{2}}{2\sigma_{r}^{2}}\right)\times \frac{1}{\sqrt{2\sigma_{\Delta \theta }^{2}\pi }}\exp\left(-\frac{\left(\Delta \theta \;-(\mathrm{atan}\left(\frac{y_{j}-y_{i}}{x_{j}-x_{i}}\right)-\theta_{i}\right){)}^{2}}{2\sigma_{\Delta \theta }^{2}}\right)$$
where *r* is the moving distance on the robot’s odometer, dist(i, j) is the distance between states *i* and *j,*
Δθ is the direction change value in the odometer, $\theta_{i}$ is the direction value of state *i* at t−1, and $\sigma_{r}$ and $\sigma_{\Delta \theta }$ represent the standard deviations for *r* and Δθ, respectively.

## 6. Results

### 6.1. Experimental Setup

The experiments were conducted in a medium-scale office building, the N1 building at the Korea Advanced Institute of Science and Technology (KAIST), Daejeon, South Korea. In the verification of the proposed methods, the robot can visit any location as much as possible to collect high-quality WRM training data. For this task, a random walker was used, and the robot was driven at a low speed of 0.1 m/s. To verify the SDCA and EVSFM proposed in this paper, a WRM was created in a part of the seventh floor of the KAIST N1 building, and the effects of each applied technology were gradually evaluated. Figure 8 shows the target area and data collection path.

The collection path of the training and test data using SLAM is shown in Figure 8b,c. The collected data and experimental environment are shown in Table 2, and the SLAM settings are shown in Table 3. In the experiment, the kNN (*k* = 1) technique was used to verify the effect of each detailed step, such as the WRM training data collection strategy and interpolation method, on the positioning accuracy.

### 6.2. WRM Training Data Verification

Figure 9 shows the results of the WRM construction while the robot builds a map in the test area. The WRM heat map visualizes the number of Wi-Fi fingerprint collections in each square area by dividing the area where the robot can move into a 50-cm × 50-cm-square area.

When the first data collection in the test area was completed, the degree of WRM collection was analyzed. At this time, to determine the recollection area of the Wi-Fi fingerprints, the robot’s data collection trajectories were analyzed, and a recollection path was planned. When the Wi-Fi fingerprint recollection path planning was completed, the robot recollected the Wi-Fi fingerprint data while navigating the waypoints along the recollection path. Comparing the collected data analysis heat map in Figure 9 and the heat map that has been recollected, the recollected data heat map is generally brighter. In other words, most of the Wi-Fi fingerprints that were not collected in the initial WRM were supplemented. The cumulative distribution function graph of the positioning accuracy before and after the recollection in Figure 9 shows that the quality of the WRM improved after the recollection. By configuring the WRM that guarantees a number of Wi-Fi collections above a certain level, the overall accuracy of Wi-Fi positioning can be improved. 

### 6.3. WRM Optimization Verification

In the WRM optimization verification, the SFM, IDW interpolation, and raw WRM (RWRM) were compared. For verification, the WRM was constructed using each method, and the positioning accuracy was measured from the constructed WRM. At this time, the number of training data was varied to check the effect of each method according to the environment.

Figure 10 shows the positioning errors when using the WRM constructed according to each method. As shown in the graph, the tendency to improve the accuracy for the number of training data was similar for all three methods. This trend continued until the number of training data was 400, and no further improvement in accuracy was observed after that. Thus, in this experimental environment, about 400 learning data are needed to build an optimized WRM. As shown in the figure, IDW and an SFM are useful for improving the quality of a WRM. The SFM technique showed a few 12.9% better performances than IDW, and the effectiveness of the SFM technique is particularly high as the number of data decreases.

### 6.4. SFM-Based Localization/Tracking Accuracy Test

In the verification of the SFM localization and tracking accuracy, the performance of the proposed SFM-based single location (SSFM) and location tracking (VSFM) are simultaneously verified using the Viterbi algorithm and SFM. The most widely used Euclidean distance-based single positioning (SE) and the positioning accuracy of the basic Viterbi algorithm and Euclidean simultaneous use (VE) were compared. To exclude the effect of improving the accuracy through an interpolation-based WRM optimization, an experiment was performed based on the WRM without an interpolation, and the accuracy of 20 test sequences was measured.

As shown in Figure 11, compared to the single-positioning SE using the Euclidean distance, the location tracking technique using the VE showed approximately a 10.6% superior performance. The SSFM showed a similar performance to the VE, despite not using tracking. In the case of VSFM tracking, the accuracy was improved by approximately 7.0% compared to the SSFM. This finding shows that, depending on the experimental environment, the SFM alone approaches the limit that can be reached by Wi-Fi positioning.

### 6.5. Integrated Localization/Tracking Accuracy Test

In this section, we verify the effect of using the SDCA learning data collection strategy, interpolation technique, and EVSFM-based positioning technique. Unlike the use of 1-NN in each module verification, the 3-NN positioning strategy was used to derive the most optimized performance. In other words, in SSFM positioning, the three most suitable locations were selected and averaged. In the case of location tracking using VSFM, the top three tracking results were averaged. In addition to SSFM and VSFM tracking, EVSFM tracking using the odometer data provided by the ROS was also verified. These combinations were compared to the basic single positioning technique (SKNN) using WRM without interpolation and 3-NN based on the Euclidean distance.

#### 6.5.1. Accuracy Test According to the Number of APs

The experiment was conducted in two ways. First, the positioning accuracy was measured by varying the number of APs used for positioning and WRM construction. Figure 12 shows the results of measuring the accuracy of SSFM, VSFM, and EVSFM using SKNN and the WRM with interpolation using the WRM (number of training data = 400) before recollection. As shown in the figure, because all the techniques use the strategy of 3-NN instead of 1-NN, a better positioning accuracy than the previous experiment was obtained. In addition, all the techniques used in the experiment showed a higher accuracy as the number of APs used for positioning increased. Overall, the accuracy of the EVSFM technique was high, and if the number of APs was sufficient, then a positioning accuracy close to 1 m could be obtained. In particular, the VSFM technique has a great effect in an environment where the number of APs is not sufficient, and when the number of APs is 10, the positioning error is reduced by approximately 40% compared to SKNN. In addition, the difference in accuracy depending on whether or not an odometer is used for location tracking is low. Thus, even if we do not use an additional odometer, we can obtain very accurate positioning results by combining WRM optimization, SFM, and the Viterbi algorithm.

Second, to show the effect of increasing the number of training data, accuracy experiments for each method were conducted after the recollection for WRM correction. Figure 13 shows the results of measuring the accuracy of each number of APs using 852 training data. The results of the experiment are similar to those shown in Figure 12, but the accuracy difference was shown in the environment with few APs. In the WRM before the recollection of the training data, the SKNN showed an accuracy of 4.1 m, as shown in Figure 12, whereas the accuracy was 3.5 m after the recollection, as shown in Figure 13. The SSFM, VSFM, and EVSFM techniques proposed in this study showed a similar degree of accuracy improvement. The improvement of accuracy through recollection tended to decrease as the number of APs increased. Thus, recollection for WRM correction has a great effect in an environment where the number of APs is insufficient.

#### 6.5.2. Comparison Test between SLAM Re-localization and Wi-Fi-Based Localization

The biggest advantage of Wi-Fi-based localization over SLAM is that the former enables fast positioning. In particular, when location recognition is required in a robot kidnapping situation, SLAM needs ample amount of time, and in some cases, it needs to move the location to search the surroundings. Moreover, Wi-Fi-based localization has the advantage that positioning is possible as soon as a Wi-Fi signal is measured. In this section, we compare the positioning delay time with the general SLAM to show the features of the proposed method in robot kidnapping situations. This experiment was verified using 10 test traces obtained twice in a total of five regions, and about 15 to 20 fingerprints were scanned in one trace.

Figure 14 shows the positioning accuracy over time obtained from nine test traces, except for one test trace that failed the SLAM re-localization. As shown in Figure 14, SLAM re-localization takes tens of seconds to determine the correct location. By contrast, Wi-Fi-based localization shows that results are immediately close to the right location. The fastest SLAM re-localization took 30 s, and the last successful, 40 s. The Wi-Fi-based localization has a positioning delay of about 0 to 3 s due to the influence of the Wi-Fi scan period. There was no significant difference between SSFM and VSFM in this experiment. The results show that it is more reasonable to use Wi-Fi technology than SLAM technology alone in the re-localization of robots.

## 7. Discussion

The main purpose of the data collection method and tracking algorithm in this study is to compensate for the degradation of the tracking accuracy by providing fusion models that can be used in the universal robot-based indoor localization system. This aim was accomplished by developing SDCA for high-quality WRM construction and EVSFM for more accurate tracking results. The proposed SDCA attempted to increase the reliability of training data construction using Wi-Fi fingerprint information, unlike previous studies that used only robot SLAM to build training data. In addition, the collected training data were analyzed, and insufficient data were supplemented to help secure high-quality training data. Moreover, unlike general research that improves one tracking algorithm, it is possible to derive more accurate positioning results by fusing the existing algorithm and the new method.

The experiments were conducted under an office building environment where the proposed algorithms could provide a precise fusion model. For WRM optimization verification, we quantified the improvement in accuracy for the common methods and SFM according to the number of training samples. In this experiment, the proposed method achieved a 12.9% improvement over the existing method. To verify the accuracy of SFM-based positioning/location tracking, the experiment was divided into SE, VE, SSFM, and VSFM components. Compared to single-positioning SE, the proposed VE showed approximately a 10.6% increased performance. In addition, it was confirmed that the proposed SSFM improved by about 7% when compared to VSFM. Finally, for verification of the integrated location recognition accuracy, the positioning accuracy of SKNN, SSFM, VSFM, and EVSFM according to the number of APs used was compared. It was confirmed that, compared to SKNN, the proposed techniques reduce the positioning error by about 40%.

In the test area, the proposed data collection method achieved 12.9% improvement, and the tracking accuracy was improved up to 40% in the 2D space during tracking. These results imply that a robot-based indoor navigation service would be possible through SDCA and EVSFM, even under harsh environments for SLAM, such as office building spaces.

By contrast, in Wi-Fi-based positioning using the characteristic of changing the signal strength due to an obstacle, such as a wall at a specific location, it is still difficult to perform positioning in an open space. The proposed algorithms, which use HMM-based tracking and Wi-Fi signals, also do not work well in open spaces. Although the proposed methods worked well in a corridor and wall environment, additional research is needed to adopt the proposed methods in various open spaces.

Meanwhile, as the WRM represents the signal environment of the corresponding indoor space, WRMs corresponding to each space must be collected afresh in a new building or space. This is a potential problem not only with this technique but, also, with all fingerprinting-based positioning techniques. Likewise, this technique uses GMapping SLAM through a robot for WRM construction; therefore, for positioning in a new space, the entire process (including data collection for SLAM, as well as WRM construction and optimization processes) has to be repeated.

In addition, the effect of environmental factors on the Wi-Fi signal strength should be considered. When using the AP signal, we set the threshold of the signal strength available for positioning to about −70 dBm, and the APs below that level were not used. Based on our experience, strong signals are dominant in determining the location, and the weaker signals increase the computational load and often cause accuracy degradation due to the noise in the received signal. A weak signal environment generally occurs when a strong AP signal is not sufficiently captured, mostly because the APs are not sufficiently distributed. The performance of the proposed scheme according to the number of APs is described with experimental result graphs in Section 6.5.1, wherein it showed up to a 40% improved performance compared to SKNN when the signal environment was poor. However, as the threshold we set is not an optimized value that can be applied to all indoor environments, we plan to look into this issue in greater depth in future studies.

## 8. Conclusions

This paper proposes SDCA and EVSFM methods for learning data collection and robot tracking. In SDCA, a WRM was created using the grid coordinates of the SLAM metric map (2D OGM) obtained from the robot equipped with the SLAM module and the Wi-Fi fingerprints collected at the same time. In the proposed EVSFM, the robot localization and tracking were calculated by combining the extended Viterbi algorithm, SFM, and odometer information under an optimized WRM in the SDCA.

Our findings confirmed that SDCA and EVSFM can improve a WRM construction and robot localization/tracking accuracy compared with existing methods. In addition, the effectiveness of the SDCA and EVSFM was verified through a detailed verification of the WRM learning data, WRM optimization, SFM localization/tracking, and integrated location recognition accuracy. In this regard, improvements are planned for the proposed method, such as an in-depth analysis of the sensors used for robot positioning and optimization of the variables. We hope that our research will help spread the scope of robotic positioning systems and widen their applications across IoT devices.

## Figures and Tables

**Figure 1 sensors-20-05182-f001:**
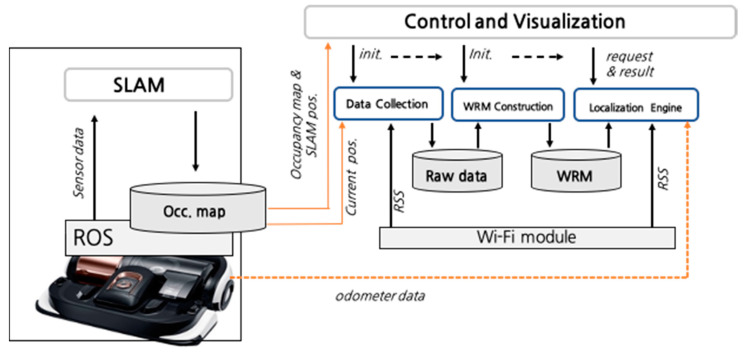
Structure of the robot localization application. SLAM: simultaneous localization and mapping, ROS: robot operating system, WRM: Wi-Fi radio map, and RSS: received signal strength.

**Figure 2 sensors-20-05182-f002:**
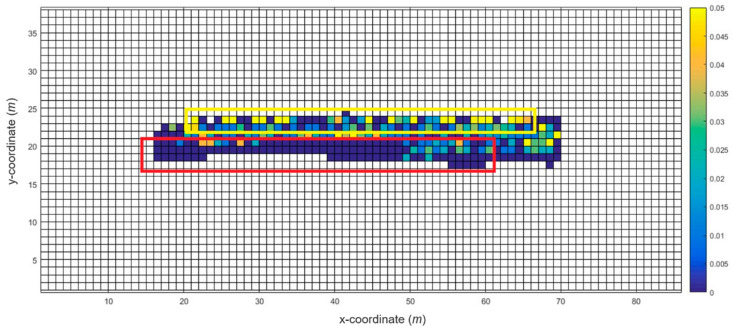
Wi-Fi fingerprint heat map example of the test bed.

**Figure 3 sensors-20-05182-f003:**
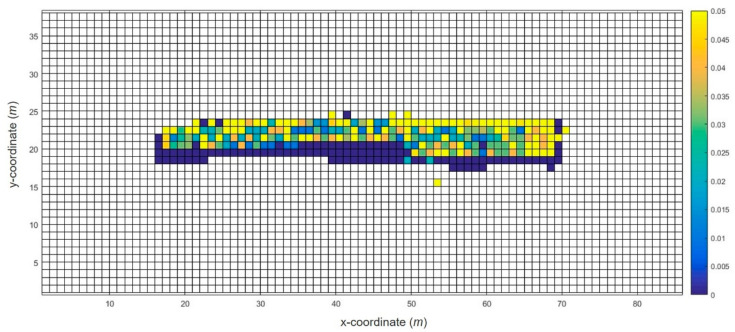
Modified WRM heat map of the test bed example.

**Figure 4 sensors-20-05182-f004:**
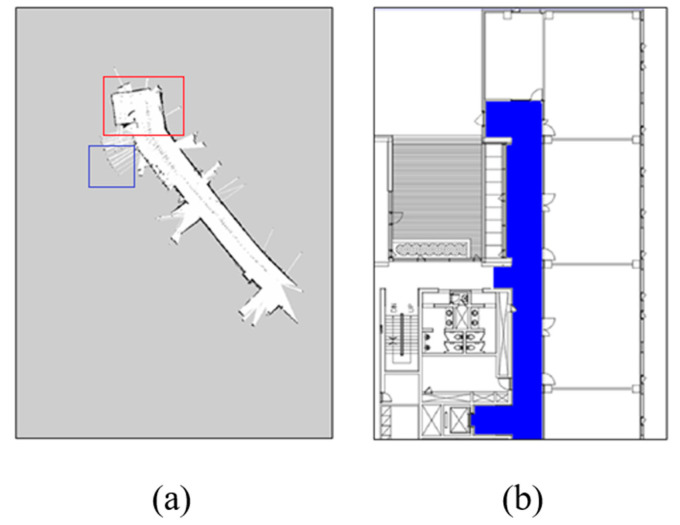
Example of collecting the occupancy grid map (OGM) of the test bed: (**a**) result of the OGM generation; (**b**) corresponding area.

**Figure 5 sensors-20-05182-f005:**
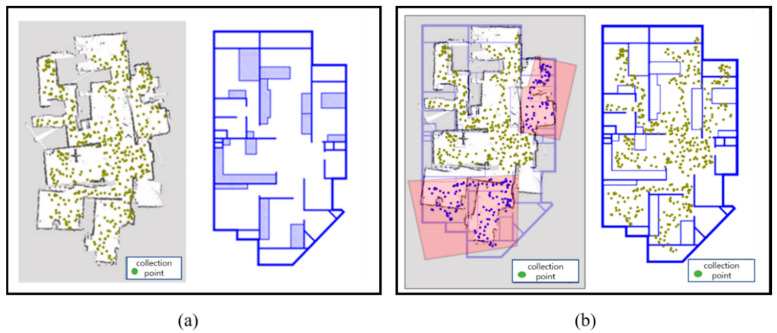
Collected WRM and image map: (**a**) before map correction and (**b**) after adopting map correction.

**Figure 6 sensors-20-05182-f006:**
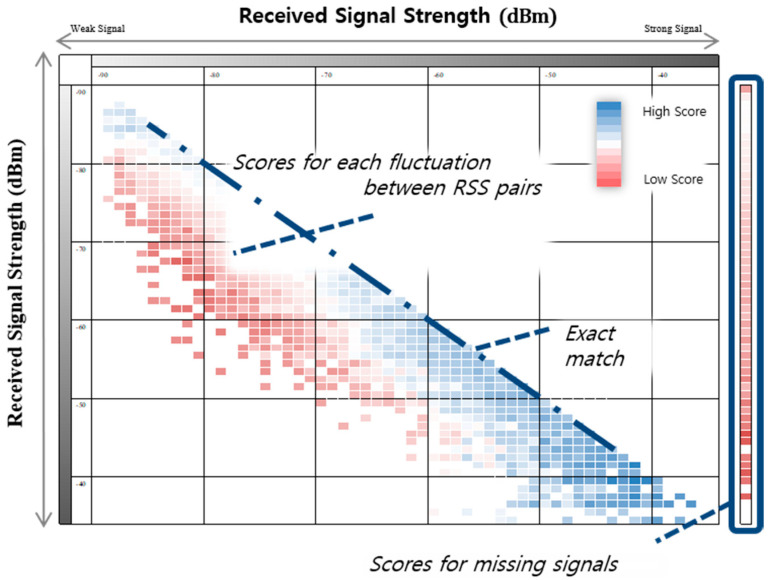
Example of a signal fluctuation matrix (SFM).

**Figure 7 sensors-20-05182-f007:**
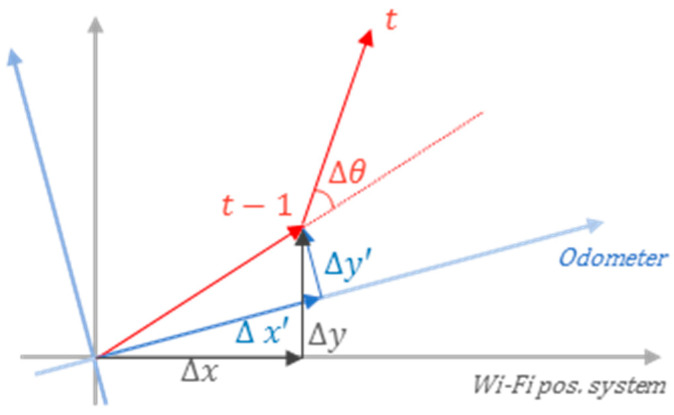
Changes in the coordinates of the odometer and Wi-Fi positioning system according to the robot movements.

**Figure 8 sensors-20-05182-f008:**
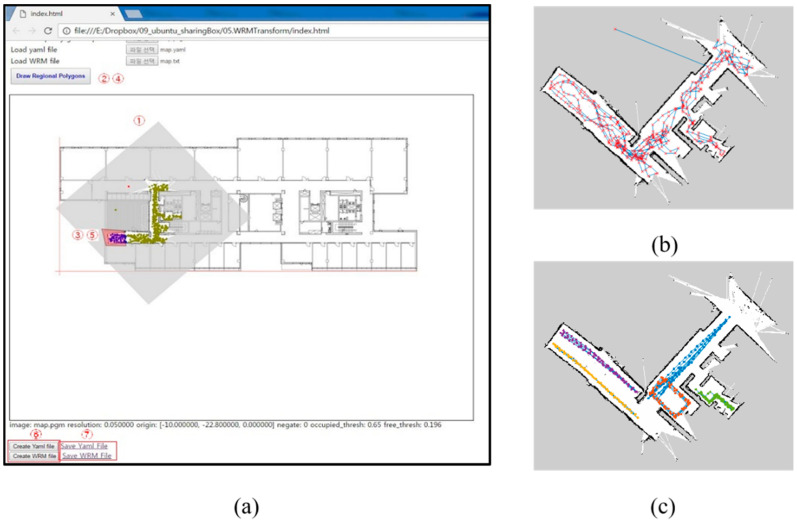
Test space in the Korea Advanced Institute of Science and Technology (KAIST) N1 building: (**a**) visualized test space in the testing tool, (**b**) training data collection path, and (**c**) test data collection path.

**Figure 9 sensors-20-05182-f009:**
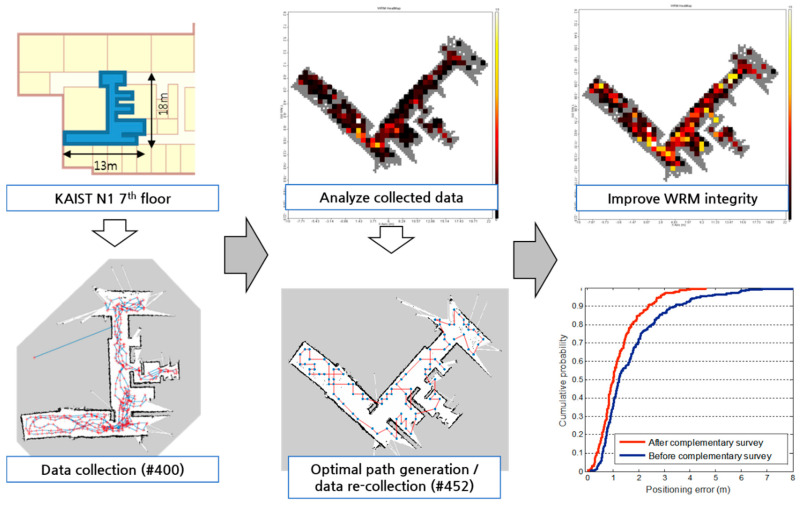
Wi-Fi fingerprint collection/recollection process of the test area.

**Figure 10 sensors-20-05182-f010:**
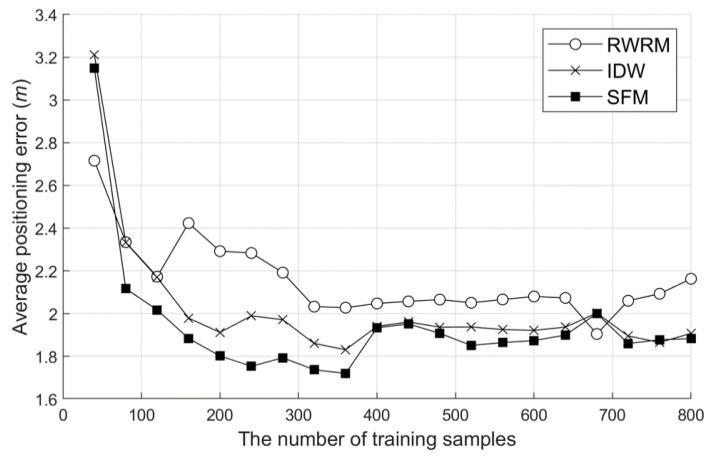
Positioning average errors according to the interpolation method. RWRM: raw WRM and IDW: inverse distance weighting.

**Figure 11 sensors-20-05182-f011:**
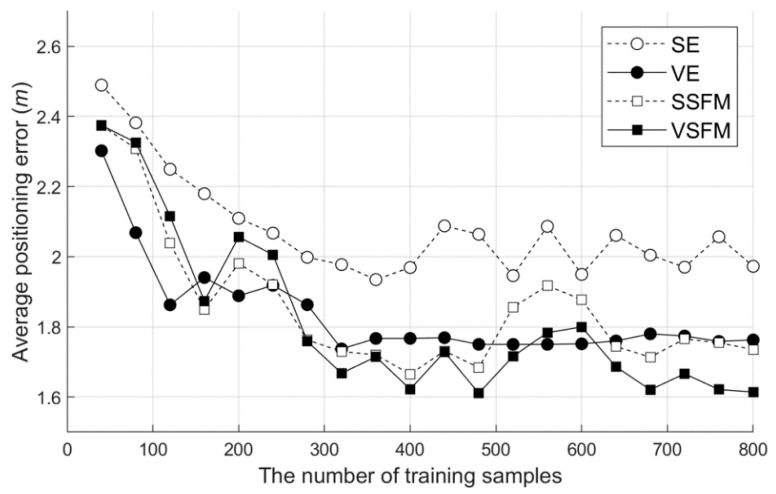
Positioning average error by the positioning algorithm. SE: Euclidean distance-based single positioning, VE: the basic Viterbi algorithm and Euclidean simultaneous use, SSFM: SFM-based single location, and VSFM: SFM-based location tracking.

**Figure 12 sensors-20-05182-f012:**
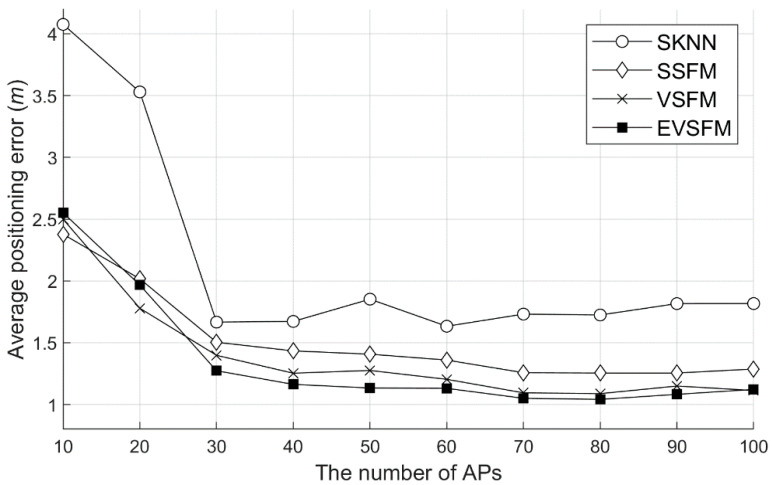
Positioning average error by the positioning algorithms (before recollection, # 400 training samples). SKNN: the basic single positioning technique and EVSFM: extended Viterbi algorithm and signal fluctuation matrix fusion tracking method.

**Figure 13 sensors-20-05182-f013:**
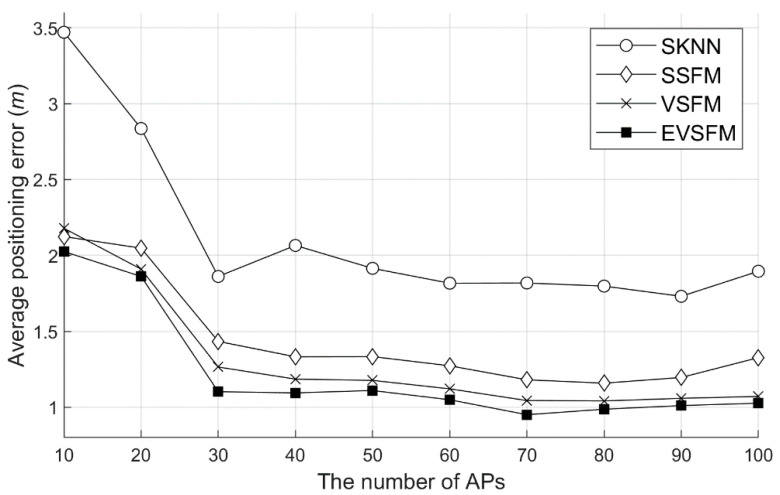
Positioning average error by the positioning algorithm (after recollection, # 852 training samples). APs: access points.

**Figure 14 sensors-20-05182-f014:**
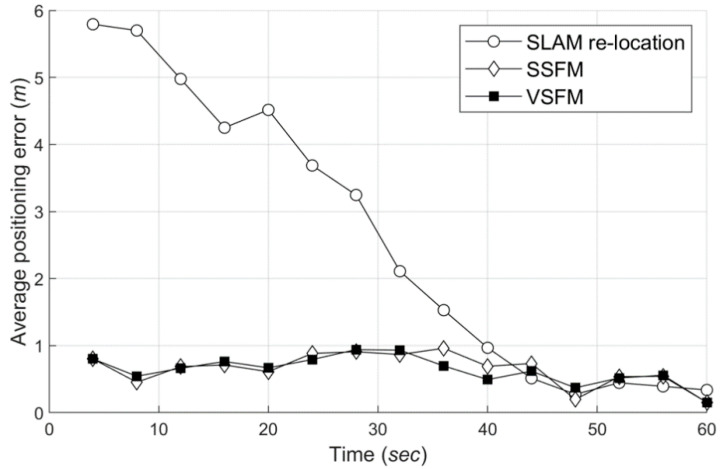
Positioning average error by the positioning algorithm (after recollection, # 852 training samples).

**Table 1 sensors-20-05182-t001:** Transformation matrix for the map correction.

Matrix Type	Moving	Rotation	Size Adjustment
**Formula**	$\left[\begin{array}{ccc} 1 & 0 & 0\\ 0 & 1 & 0\\ dx & dy & 1 \end{array}\right]$	$\left[\begin{array}{ccc} cos\theta & sin\theta & 0\\ -sin\theta & cos\theta & 0\\ 0 & 0 & 1 \end{array}\right]$	$\left[\begin{array}{ccc} sx & 0 & 0\\ 0 & sy & 0\\ 0 & 0 & 1 \end{array}\right]$

**Table 2 sensors-20-05182-t002:** Experimental environment and data collection specifications. APs: access points.

**Size of area**	89 $m^{2}$
**# Signal measurements (training data)**	852 (first trial: 400; second trial: 452)
**Wi-Fi scanning interval**	2–3 s
**Time required**	2.5 h
**Measurement density**	3.64/$m^{2}$
**Total # APs detected**	118
**Avg # APs in a measurement**	37.86
**# Test data**	557

**Table 3 sensors-20-05182-t003:** Simultaneous localization and mapping (SLAM) settings. OS: operating system.

**Robot**	Kobuki (Yujin Robot, Korea)
**OS**	Ubuntu 14.04 LTS/ROS Indigo
**Laptop**	Intel Celeron N3050/DDR3L 4G (Samsung, Korea)
**Laser**	URG-04LX (HOKUTO, Japan)

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
