# Peer review of "Fusion of the SLAM with Wi-Fi-Based Positioning Methods for Mobile Robot-Based Learning Data Collection, Localization, and Tracking in Indoor Spaces"

_sensors, 2020, doi:10.3390/s20185182_

Round 1

Reviewer 1 Report

The paper presents an interesting subject. It is well written. Some aspects must be more detailed:

  • In particular, when location recognition is required in a robot kidnapping situation, SLAM needs ample amount of time, and in some cases, it needs to move the location to search the surroundings. -> it will be better if some measurements will be added about "ample amount of time" - please explain what are these values.

  • what will be the results in case of a low WiFi signal
  • what are steps that are needed in case of applying the proposed algorithms in a new building / space

Reviewer 2 Report

The paper is interesting and the proposed solution improve significantly the performance  of the common approach

Moreover some revision are suggested 

Section 2. Related work.

The section seems a summary of most important work related with the paper. Critical analysis of the cited paper is completely missing.

Section 3. Methods

The section is too descriptive. Introduction of a much more analytical description of the method is suggested

Figure 2 and Figure 3.  

X and Y axis label are missing. No sufficient description is included in text.

Section 4. WRM Construction using SDCA

In 4.2.1 Subsection Interpolation concept is introduced but no description is supplied

Section 7. Discussion

The section has to be strengthened and should refer further to section 6 where the results are described

Round 2

Reviewer 1 Report

Since all my comments were addressed, I recommend to publish the paper.

Reviewer 2 Report

The author revision improve the quality of the paper.

The peper is now much more intelligible.

The paper may be accepted in present form 

Thanks for fixing citation problems